# Therapeutic Strategies to Prevent the Recurrence of Nasal Polyps after Surgical Treatment: An Update and In Vitro Study on Growth Inhibition of Fibroblasts

**DOI:** 10.3390/jcm12082841

**Published:** 2023-04-13

**Authors:** Angela Rizzi, Luca Gammeri, Raffaele Cordiano, Mariagrazia Valentini, Michele Centrone, Sabino Marrone, Riccardo Inchingolo, Franziska Michaela Lohmeyer, Carlo Cavaliere, Francesco Ria, Gabriella Cadoni, Sebastiano Gangemi, Eleonora Nucera

**Affiliations:** 1UOSD Allergologia e Immunologia Clinica, Dipartimento Scienze Mediche e Chirurgiche, Fondazione Policlinico Universitario A. Gemelli IRCCS, 00168 Rome, Italy; angela.rizzi@policlinicogemelli.it (A.R.); michelecent87@hotmail.it (M.C.); eleonora.nucera@policlinicogemelli.it (E.N.); 2Department of Clinical and Experimental Medicine, School and Operative Unit of Allergy and Clinical Immunology, University of Messina, 98125 Messina, Italy; lucagammeri@outlook.com (L.G.); raffaelecordiano@gmail.com (R.C.); gangemis@unime.it (S.G.); 3Department of Obstetrics and Gynecology, Catholic University of the Sacred Heart, 00168 Rome, Italy; mariagrazia.valentini@policlinicogemelli.it; 4Division of Gynecologic Oncology, Fondazione Policlinico Universitario A. Gemelli IRCCS, 00168 Rome, Italy; 5Otorhinolaryngology Unit, Head and Neck Department, Fondazione Policlinico Universitario A. Gemelli IRCCS, 00168 Rome, Italy; sabinomarrone@gmail.com (S.M.); gabriella.cadoni@unicatt.it (G.C.); 6UOC Pneumologia, Dipartimento Neuroscienze, Organi di Senso e Torace, Fondazione Policlinico Universitario A. Gemelli IRCCS, 00168 Rome, Italy; 7Direzione Scientifica, Fondazione Policlinico Universitario A. Gemelli IRCCS, 00168 Rome, Italy; franziska1.lohmeyer@gmail.com; 8Department of Sense Organs, Sapienza University, 00185 Rome, Italy; carlo.cavaliere@uniroma1.it; 9Section of General Pathology, Department of Translational Medicine and Surgery, Università Cattolica del Sacro Cuore, Largo Francesco Vito 1, 00168 Rome, Italy; francesco.ria@unicatt.it; 10Department Laboratory and Infectious Diseases Sciences, Fondazione Policlinico Universitario A. Gemelli IRCCS, Largo Agostino Gemelli 1-8, 00168 Rome, Italy; 11Otolaryngology Institute, Università Cattolica del Sacro Cuore, 00168 Rome, Italy; 12Medicina e Chirurgia Traslazionale, Università Cattolica del Sacro Cuore, 00168 Rome, Italy

**Keywords:** chronic rhinosinusitis with nasal polyps, surgical polypectomy, intranasal corticosteroids, non-steroidal anti-inflammatory drugs, lysine-acetylsalicylate, diclofenac, ketoprofen, fibroblast, proliferation, biological therapies

## Abstract

Chronic rhinosinusitis with nasal polyps (CRSwNP) is the most bothersome phenotype of chronic rhinosinusitis, which is typically characterized by a Type 2 inflammatory reaction, comorbidities and high rates of nasal polyp recurrence, causing severe impact on quality of life. Nasal polyp recurrence rates, defined as the number of patients undergoing revision endoscopic sinus surgery, are 20% within a 5 year period after surgery. The cornerstone of CRSwNP management consists of anti-inflammatory treatment with local corticosteroids. We performed a literature review regarding the therapeutic strategies used to prevent nasal polyp recurrence after surgical treatment. Finally, we report an in vitro study evaluating the efficacy of lysine–acetylsalicylic acid and other non-steroidal anti-inflammatory drugs (ketoprofen and diclofenac) on the proliferation of fibroblasts, obtained from nasal polyp tissue samples. Our study demonstrates that diclofenac, even more so than lysine–acetylsalicylic acid, significantly inhibits fibroblast proliferation and could be considered a valid therapeutic strategy in preventing CRSwNP recurrence.

## 1. Introduction

Chronic rhinosinusitis (CRS) includes a heterogeneous group of conditions with varying pathophysiology. To date, two main subgroups have been identified: chronic rhinosinusitis with nasal polyps (CRSwNP) and CRS without nasal polyps [1,2]. CRSwNP represents the clinical expression of a persistent local inflammation that causes mucosal degeneration.

The pathogenetic mechanism of nasal polyps is poorly understood. Most theories consider polyps as the ultimate manifestation of chronic inflammation. However, few da-ta are available on epithelial changes and their relationship to free radical damage [3]. From a histological point of view, polyps are benign translucent outgrowths of edematous inflammatory tissue, which is, in its stromal component, mainly composed of fibroblasts.

Over time, many theories have been proposed to elucidate the nature and formation of polyps. They have been characterized as adenomas, fibroids, mucosal exudates, cystic dilatations of excretory ducts, or swellings due to obstruction of glands or lymphatics. Alternative theories hypothesized that lymphangitis from recurrent infection or glandular hyperplasia concurred in the genesis of the nasal polyp [4]. The majority of patients with CRSwNP have a Type 2 pattern of inflammation [5,6] characterized by eosinophilia and high levels of interleukin-4, interleukin-5, and interleukin-13 cytokines. Eosinophilic and M2 macrophage infiltration was associated with a worse level of edema and an abundance of fibronectin deposits in CRSwNPs.

Tissue swelling is evidenced by the presence of plasma proteins and prominent edema, with increased angiogenesis and vascular endothelial cell growth factor expression. Increased glandular activity, secretions, globular cell number and activity were also noted in nasal polyp tissue. Epithelial alterations contribute to polyp formation. Increased levels of matrix metalloproteinase have been hypothesized to promote polyp formation through extracellular matrix degradation and subsequent repair events. Indeed, extensive collagen deposition is present in the stalk of the early polyp. Epithelial loss occurs in polyp formations due to the deficiency of junction proteins (e.g., E-cadherin, zonula occludens-1 (ZO-1), and occlusion). At the same time, vascular leakage can lead to thrombin activation and the cleavage of plasma fibrinogen in polyp tissue, in which fibrinolytic capacity is reduced [7].

The association between nasal polyps and allergic rhinitis remains unclear. Nasal polyps are reported to be less common in patients with allergic rhinitis and childhood-onset allergic asthma [8]. According to current literature, the role of allergy in CRSwNP and CRS without nasal poly continues to be controversial, with insufficient evidence.

The patient’s symptoms in CRSwNP depend on the size of the polyps. Small polyps may not cause symptoms; they can be identified during a routine examination if they are anterior to the middle turbinate. However, a flexible fiberscope is required when the polyps are located posteriorly. Flexible fibroscopy is the best method to examine the nasal cavity, evaluate the nasal anatomy fully, and determine the exact location of nasal polyps.

Clinically silent in initial stages, problems related to these formations are due to nasal obstruction and consequent complications. Nasal obstruction leads to recurrent inflammation of the upper and lower airways (rhinosinusitis, otitis media, pharyngotonsillitis-litis, and tracheobronchitis). In addition, nasal obstruction is usually already associated with mucous secretion due to insufficient drainage of the paranasal sinuses, with the possibility of bacterial superinfection. Another characteristic symptom, which often leads the patient to consult the doctor in the initial stage of the disease, is the reduction of olfactory sensitivity, due to the mechanical obstruction caused by the polyps.

Computerized tomography in axial and coronal planes is the only exam that can be used to accurately diagnose the polyps’ extension. The increasingly precise, rapid, and less-invasive imaging techniques are now fundamental in staging the disease and identifying associated anatomical anomalies or any complications to the occurrence of polyps [2].

Medical treatment options for patients with CRSwNP remain limited. According to recent guidelines, topical corticosteroids are recommended as an initial medical therapy for affected patients, although various therapeutic strategies (local and systemic) have been tested. Numerous studies have shown that intranasal glucocorticoids are more effective than placebo in reducing symptoms, including nasal blockage, rhinorrhea, loss of smell, and polyp growth [9]. Some adverse events, such as nasal irritation and epistaxis, have been reported, especially with their long-term use. However, systemic effects are rare considering the minimal quantities absorbed (less than 1%), in particular, by second-generation compounds (mometasone and fluticasone). Monoclonal antibodies targeting the inflammatory pathway have been suggested as another therapy for CRSwNP [2,10]. Dupilumab (anti-interleukin-4 and anti-interleukin-13 receptors) was the first monoclonal antibody approved to treat CRSwNP [11]. The use of other biological therapies, such as omalizumab (anti-IgE) and mepolizumab (anti-interleukin-5), significantly reduced both symptom scores and polyp size [12,13]. Long-term safety, cost-effectiveness, the risk of anaphylaxis, and the use of subcutaneous injection are limiting factors in the current widespread application of such treatments.

Endoscopic sinus surgery is usually reserved for patients who have not responded to medical therapy. Although polyps do not show the characteristics of malignant neoformations, one of the fundamental characteristics of nasal polyps is that they demonstrate a high tendency to relapse, even in the case of an appropriate surgical treatment of the nasal cavities. Regardless of the surgical technique used (macro-surgical, microsurgical, or endoscopic), the frequency of recurrence is around 30% a few years after surgical treatment [14].

What has been reported so far highlights the social weight of the pathology, which, although not life-threatening, can limit quality of life. The main critical points are sleep disorders, anosmia, prolonged medical treatment, and repeated surgical interventions [15,16].

Topical glucocorticoids are often used after endoscopic surgery to prevent recurrence. Lysine–acetylsalicylic acid (LAS), similar to other non-steroidal anti-inflammatory drugs (NSAIDs), interferes with various intracellular functions and with a series of biochemical events that, alone or in combination, can influence cell growth [17].

We performed an update about the therapeutic strategies used to prevent the recurrence of nasal polyps after surgical treatment. Furthermore, we focused briefly on the potential role of biological therapy. Finally, based on previous in vitro studies, our study aims to evaluate the in vitro efficacy of LAS and other NSAIDs on fibroblast growth.

## 2. Update of Molecules Used to Prevent Nasal Polyp Recurrences

### 2.1. Intranasal Corticosteroids (INCs)

Intranasal corticosteroids (INCs) are the first-line therapeutical approach to CRSwNP. These molecules have an anti-congestion effect on the nasal mucosa and cause a reduction in polyp size, increasing nasal airflow [18]. The safety of INCs is well known. Many clinical trials have demonstrated the efficacy and safety profile of corticosteroids, such as mometasone furoate (MF), fluticasone furoate (FF), and fluticasone propionate (FP) [19]. INCs used in the postoperative phase allow good control of symptoms, and some authors described a reduction of polyp recurrence in the first year after functional endoscopic sinus surgery (FESS) [20]. In addition to anti-inflammatory and anti-edema effects, INCs could inhibit fibroblast proliferation (Figure 1).

Yariktas et al. demonstrated a reduction of basic fibroblast growth factor in the nasal mucosa of patients with nasal polyposis after INC treatment (mometasone furoate, 200 µg, once daily) [21]. Yu et al. demonstrated the role of glucocorticoids in inhibiting tissue remodeling in a clinical trial on 30 patients with CRSwNP after endoscopic surgery. The thickness of the basement membrane after three months of therapy was significantly reduced in the glucocorticoids group compared to the control group (*p* < 0.05) [22].

The first studies about INC use for polyp recurrence prevention were published in the 1980s. Virolainen and Puhakka demonstrated the efficacy of beclomethasone dipropionate in preventing the recurrence of nasal polyps. In their work, one year after surgery, polyps were absent in 54% of the patients treated with beclomethasone dipropionate. In the control group, polyps were absent in 13% of the patients [23]. These differences compared to the untreated subjects were observable after six months [24]. Treatment with flunisolide has been shown to reduce the risk of polyp recurrence in the first year after polypectomy [25,26]. Hartwig et al. demonstrated the safety and efficacy of budesonide (400 µg/daily) in preventing nasal polyps after sinus surgery [27]. A high dose of INC could be more effective as compared to a low dose of INC [28].

About that Kang et al. evaluated the recurrence of polyposis in 32 patients subjected to FESS. After 12 months, polyps recurred in 7.1% of patients treated with high-dose corticosteroids (triamcinolone acetonide–soaked gauze packing). In the group treated with low-dose corticosteroids, polyps recurred in 44% of patients [28].

However, in a prospective, double-blind, randomized study, triamcinolone acetonide had no significant efficacy compared to placebo, in patients with concomitant acetylsalicylic acid (ASA) intolerance [29].

Nevertheless, Seiberling and her group did not observe a substantial difference be-tween high-dose INC (dexamethasone) and low-dose INC such as FP [30].

Mometasone furoate has been shown to be effective in prolonging recurrence-free time [31]. In a randomized, double-blind, placebo-controlled, multicenter trial on 159 subjects, Stjärne et al. demonstrated that mometasone (200 µg, once daily) was superior to placebo.

Dijkstra et al. did not observe a reduction in the recurrence rate of nasal polyps in patients treated with fluticasone propionate after FESS. The recurrence rate of patients treated with intranasal FP after FESS was similar to the placebo group [32].

### 2.2. Acetylsalicylic Acid and Other NSAIDs

One of the main “alternative” pharmaceutical compounds used to reduce the recurrence of nasal polyps following surgical treatment is acetylsalicylic acid (ASA, aspirin). Since it is insoluble in liquids, a lysine molecule needs to be added to the acetylsalicylic acid to allow the use of lysine aspirin or lysine–acetylsalicylate (LAS) for inhalation or nasal challenge tests.

Lysine aspirin or lysine–acetylsalicylate can be used in patients with and without a history of aspirin sensitivity. Additionally, a condition known as the Samter triad, or aspirin illness, is defined as the coexistence of a nasal polyp, aspirin sensibility, and concurrent asthma. In aspirin-sensitive individuals, a repeated intake of ASA causes no adverse reaction during a “refractory period”. The presence of this period after a nasal challenge prompted researchers to study its topical effect on nasal polyp growth in ASA-sensitive patients [33].

From a pathophysiologic perspective, ASA affects inflammation by inhibiting cyclo-oxygenases (COX-1 and COX-2), which causes a shift in the metabolism of arachidonic acid toward the lipoxygenase pathway; this leads to an increase in leukotriene synthesis, and a decrease in prostaglandin E2. In ASA-sensitive patients, these traits potentially precipitate asthma attacks [34].

In an in vitro study, a non-specific, dose-dependent antiproliferative effect of aspirin on nasal polyps and normal skin fibroblast proliferation was revealed [35]. A report by Patriarca et al. [36] evaluated the effect of topical LAS (2000 µg weekly) on the recurrence of polyps after surgical therapy. In the 24 month study, 28 patients with ASA intolerance were enrolled (including 20 with ASA-triad), as well as 15 ASA-tolerant patients (control group). Compared to 81% of aspirin-sensitive controls, nasal polyp recurrences have been observed in 32% of treated patients. Additionally, in contrast to 67% of controls, no polyps recurred in the aspirin-tolerant patient group. Another work obtained similar results, although the results were less significant. A group of ASA-tolerant patients with nasal polyps (*n* = 20) was treated with topical LAS (2 mg weekly) unilaterally, using the other side of the nose as a control. The delayed recurrence of polyps was noted in all 18 patients, of which eight remained symptom-free after 15 months [37]. In further two long-term studies, Nucera et al. [38] assessed the effect of intranasal LAS in aspirin-sensitive and aspirin-tolerant patients with nasal polyps compared with controls. In the six-year follow-up study, 76 patients (38 with ASA sensitivity), who underwent surgical polypectomy, were treated with the equivalent of 4 mg of aspirin instilled six times a week. The recurrence rate of polyps compared with controls was 6.9% vs. 51.3% in the first year 1, 44.9% vs. 84.8% after three years, and 65% vs. 93.5% after six years. Moreover, no differences between the ASA-sensitive and ASA-tolerant patients were noted. In the second study 49 patients, who were previously treated with medical polypectomy, were enrolled, and they underwent topical applications of aspirin, as described above. At the three-year follow-up, the cumulative percentage of patients needing surgery was 32% for the LAS-treated group compared with 85% for the controls.

In contrast, a double-blinded, placebo-controlled, crossover trial [39] conducted in twenty ASA-sensitive patients did not show a significant clinical benefit on polyp growth or nasal inspiratory flow rate after topical treatment with the equivalent of 16 mg of aspirin every 48 h for six months.

Finally, in a research published by Ogata et al. [40], 13 patients with asthma and nasal polyps were included, all but one of whom previously underwent endoscopic polypectomy. LAS was applied topically for three months in doses equal to 37.8 mg of aspirin per day. When data were compared to measurements made at the start of the trial, it was found that nasal airflow, nasal nitric oxide levels, and polyp size all improved.

Other NSAIDs have been studied mainly in vitro. Ostwald et al. [41] used 12 molecules on fibroblasts from nasal polyp tissue at a concentration of 30 µg/mL. Corticosteroids achieved the strongest reduction in fibroblast growth, while NSAIDs, such as diclofenac and piroxicam, showed an intermediate-level effect. In another report [42], the selective COX-2 inhibitor, rofecoxib, at a concentration of 10,000 nM, strongly inhibited fibroblast proliferation, while nimesulide and ibuprofen showed no effect.

### 2.3. Furosemide

Other pharmacological molecules have been topically used to prevent polyp recurrence. Based on pathophysiological mechanisms, some clinical trials were performed using furosemide as an anti-edematous molecule. One of the key elements in polyp development is the edematous infiltrate [43], which grows due to inflammatory cells and their related cytokine (TNF, IL-1) and chemokine actions. These signaling molecules result in eosinophil persistence in the lamina propria of the nasal polyp, where its primary effectors, the major basic proteins, alter the ionic flux of the respiratory cell’s surface. In particular, the sodium net flux increases with a consequently augmented water absorption in the cells, resulting in edema formation [44].

The rationale for using topical furosemide in polyp recurrence is primarily based on the inhibition of a sodium–potassium–chloride symporter isoform (NKCC2) present on the basolateral membrane of nasal polyp epithelial cells [45,46]. This decreases sodium, chloride, and water absorption that leaves the interstice and reaches the surface of the mucosa.

In two different works, Passali et al. analyzed the effects of furosemide in preventing post-surgical relapses of nasal polyps. In an initial randomized clinical trial [47], 64 patients out of a cohort of 104 patients were treated with topical administrations of furosemide after surgical treatment. After a six-month endoscopic control, no relapse was reported in patients treated with furosemide (4 cases vs. 12 cases). In another work [44], the efficacy in preventing polyp formation of topical furosemide was compared with a topical corticosteroid (MF) and placebo. One hundred seventy patients were enrolled and divided into three groups (97 patients treated with furosemide, 33 treated with mometasone, and 40 treated with placebo). The results showed a comparative efficacy of furosemide and MF, with similar rates of polyp recurrence in the two groups (17.5% vs. 24.2%) compared with placebo (30%). In addition, regarding the severity of relapses, Stage III polyps were found in 11.8% of patients treated with furosemide and in 12.5% of patients treated with MF, compared with 66.7% of patients in the control group. In conclusion, in this study, furosemide was as effective as MF in preventing relapse. Moreover, patients who received no treatment experienced more severe degrees of relapse.

In a more recent, triple-blind clinical trial, Hashemian et al. [48] evaluated the severity of polyps, based on computed tomographic (CT) scans, using paranasal sinuses’ scoring methods, the Sino Nasal Outcome Test (SNOT-22), visual analog scale (VAS) and endoscopic grading using the Meltzer score, in 84 patients with CRSwNP who did not respond to medical treatment and were candidates for surgery. A substantial reduction in polyp grading was observed in the furosemide group based on endoscopic grading, SNOT-2, and VAS score [48].

### 2.4. Antibiotics and Antifungal Drugs

In the literature, evidence about the use of antibiotics or antifungal molecules to prevent nasal polyp relapse can be found. The first studies on antibiotic use evaluated the efficacy of intranasal clarithromycin administration. In vitro studies demonstrated that this macrolide inhibits the production of pro-inflammatory cytokines, such as IL-5, IL-8, and GM-CSF [49].

The anti-inflammatory effects of macrolides could be due to their ability to inhibit the activation of the transcriptional factor NF-κB [50].

Piskunov and Bobacheva randomized 37 patients to two therapeutic strategies after endoscopic polyposinusotomy. The first group was treated with 250 mg of intranasal clarithromycin and INC for three months; the second group received INC for three months. The authors observed that clarithromycin at low doses caused significant stabilization of chronic polypoid rhinosinusitis remission and prevented the development of relapses 66% of patients. This study demonstrated the efficacy of the combined therapy compared to the INC alone and the safety of intranasal clarithromycin [51].

Subsequently, other works confirmed the usefulness of macrolides in preventing the early recurrence of nasal polyps after FESS. In patients with CRSwNP and surgery failure, low-dose clarithromycin treatment (250 mg once daily) for 3–6 months could prevent relapse of nasal polyps [52,53].

Furthermore, studies about the use of mitomycin C were conducted. Mitomycin C is an antineoplastic used in ophthalmologic and sinus surgery to prevent synechiae formation [54].

This antibiotic inhibits GM-CSF synthesis with a reduction of IL-5 secretion [55], and increases eosinophil apoptosis [56].

De Castro et al. studied the nasal mucosa of patients treated with topic mitomycin C after sinus surgery, evaluating the gene expression of IL-4, IL-5, IL-10, IL-13, chemokine (C-C motif) ligand 5 (CCL5), CCL24, colony-stimulating factor 2 (CSF2), transforming growth factor beta 1 (TGFB1), tumor necrosis factor-alpha (TNF-alpha), and beta-actin by quantitative PCR. They found that topical application of this drug induced a down-regulation of IL-5, CCL, CCL24, TNF-alpha, and CSF2. Therefore, De Castro et al. concluded that mitomycin C promotes the restoration of the normal microenvironment of the nasal mucosa; this could have applications in the prevention of recurrent eosinophilic nasal polyposis [57].

Some authors have also considered using antifungals to treat symptomatic CRS to prevent relapse of nasal polyps. In fact, in the late 1990s, evidence about the presence of mycotic microorganisms in a large percentage of subjects with CRS was found. Therefore, some authors started a series of studies about intranasal amphotericin B. Nevertheless, Lupa and Amedee reviewed studies in the literature and demonstrated the lack of utility of amphotericin B to treat patients with CRS [58].

In a prospective randomized placebo-controlled trial involving 33 patients, Gerlinger et al. [59] evaluated the effects of amphotericin B to prevent nasal polyp recurrence. Results have shown that administering intranasal amphotericin B after endoscopic surgery does not produce benefits compared to placebo [59].

### 2.5. Other Therapeutical Strategies

Many other molecules have been studied for the post-surgical treatment of nasal polyposis and the prevention of recurrences. Among these, antileukotrienes have been proposed for the post-surgical treatment of aspirin-exacerbated respiratory disease (AERD) patients. These patients’ mucosal tissue overexpresses cysteinyl leukotrienes (CysLTs) receptor and leukotriene C4 synthase (LTC4S) [60].

Dysregulation of LTC4 leads to an alteration in the arachidonic acid metabolisms and the overproduction of CysLTs. CysLTs have a role in eosinophilic inflammation, exerting pro-inflammatory and pro-fibrotic effects [61]. The use of antileukotrienes after endoscopic surgery showed similar results to INC therapy [62]. From a literature review published in June 2010 emerged a class IB recommendation for the use of antileukotrienes in patients undergoing sinus surgery [61].

Vuralkan et al. [62] compared montelukast 10 mg per day and a nasal spray of 400 mg MF, administered twice daily. Fifty patients were randomized into two groups after endoscopic sinus surgery and treated for six months. Montelukast and INC demonstrated similar results in symptom control, but MF showed a slightly lower recurrence rate [62].

The mechanisms of action of the main molecules used in the prevention of post-surgical recurrence of nasal polyposis are shown in Figure 2.

Some authors evaluated the potential use of capsaicin. Sensory C-fibers in the nasal mucosa contain calcitonin gene-related peptide; this peptide has a vasodilator effect and may play a role in the vascular component of inflammation [63].

In nasal mucosa, increased levels of calcitonin gene-related peptide positively correlate with inflammatory cell infiltrate and symptom severity [64].

Zheng et al. [65] evaluated the efficacy of intranasal capsaicin application in preventing nasal polyp recurrence after sinus surgery. Patients treated with five intranasal capsaicin applications showed a significant reduction in the rate of polyp recurrences; capsaicin also demonstrated a good safety profile [65].

Isotonic and hypertonic solutions could play an important role. Saline douching in the post-surgery period could have an anti-inflammatory effect, reducing nasal discharge and edema [66]. The hypertonic solution was superior to the isotonic solution in symptom control [67].

Portenko and Dobrynin [68] studied the potential application of transcranial electrostimulation to reduce polyp recurrence in a small cohort of 23 patients with CRSwNP. They obtained a reduction in early recurrence (4.3%) in a follow-up period of 1.5–3 years [68].

Finally, an approach to the disease with herbal medicine could be considered. The anti-inflammatory effects of many herbal preparations are well known; there is various evidence on the use of decoctions to treat inflammatory skin diseases, such as urticaria and respiratory diseases (asthma, rhinitis) [69,70].

In literature, a case report about a patient with recurrent nasal polyposis treated with an herbal decoction and acupuncture is reported; this patient did not show polyp recurrence during three and half years of treatment [71].

Chang et al. used licorice (glycyrrhiza glabra) extract to treat CRSwNP through nasal irrigation and obtained a reduction of polyp size. In addition, through inhibition of the MAPK/ERK-1/2 signaling pathway, glycyrrhiza glabra could attenuate fibroblast differentiation, extracellular matrix production, and cell migration [72].

Nevertheless, studies on the use of herbal medicine to prevent nasal polyp recurrence require further insights.

### 2.6. Biological Therapies

According to the European Forum for Research and Education in Allergy and Airway Diseases (EUFOREA), patients with severe CRSwNP and comorbid asthma or those whose blood eosinophil count is higher than 300 cells/mL are more likely to exhibit Type 2 inflammation and may, therefore, benefit from Type 2 biologic therapies, such as anti–IL-4/receptor alpha and anti–IL-5/receptor alpha molecules [73].

Dupilumab is a humanized monoclonal antibody targeting the IL-4 receptor alpha chain (IL-4Rα). The alpha chain is a subunit of the IL-4R complex, a heterodimer that mediates the action of Il-4 and IL-13. The axis between IL-4/IL-13 and IL-4R promotes Th2 inflammation [74].

Dupilumab is the first monoclonal antibody to be approved in the US and Europe for the treatment of not-well-controlled CRSwNP [75].

Two phase III multicenter, randomized, double-blind, placebo-controlled trials, LIBERTY NP SINUS-24 and LIBERTY NP SINUS-52, demonstrated the efficacy of this treatment in adult patients with severe CRSwNP. Dupilumab, in addition to standard daily therapy, was well tolerated and led to a reduction of polyp size and severity of symptoms [76]. The efficacy and safety were evaluated and also demonstrated in real-world studies [77].

A humanized monoclonal antibody called mepolizumab binds to and deactivates IL-5, the main cytokine in charge of eosinophil proliferation, activation, and survival [78]. Mepolizumab has received approval for the treatment of various eosinophilic disorders in numerous countries across the world. These include CRSwNP, eosinophilic granulomatosis with polyangiitis, hyper eosinophilic syndrome, and severe eosinophilic asthma [79]. Mepolizumab reduced nasal polyp size, improved nasal obstruction symptoms, decreased the need for actual nasal surgery, decreased the use of systemic corticosteroids, improved sinonasal symptoms in patients with severe CRSwNP, and had an acceptable safety profile, based on the phase III SYNAPSE trial [80]. Furthermore, the same study also demonstrated its effectiveness in treating nasal obstruction among patients with high basal eosinophil counts [80]. A recent evaluation of SYNAPSE patients (grouped by comorbid asthma, AERD, and baseline blood eosinophil count) showed that patients with baseline blood eosinophils count >150 cells/mL versus <150 cells/mL, and those with ≥300 cells/mL versus <300 cells/mL tended to have better therapeutic benefits; this confirms that eosinophils are an appropriate and effective target in severe CRSwNP [81].

A recent multicentric real-life study on severe asthmatic patients treated with mepolizumab, with comorbid CRSwNP, showed an improvement of CRSwNP outcomes (SNOT-22 score, nasal polyp score and blood eosinophil count) in an Italian cohort of forty-three patients [82].

Finally, biological therapy for nasal polyps can also act on IgE. IgE may play a role in the inflammatory microenvironment of the nasal polyp, which is confirmed by the elevated polyclonal local IgE production in some patients with nasal polyposis [83].

Omalizumab is a humanized anti-IgE antibody that binds to the IgE and prevents IgE binding to the high-affinity receptors (FcεRI) [84]. This monoclonal antibody was found to be effective and safe in the treatment of nasal polyposis [85].

In two Phase III trials, POLYP-1 and POLYP-2, omalizumab was superior to placebo in patients with or without previous FESS. In addition, efficacy was demonstrated regardless of eosinophil levels, as well as the concomitant presence of asthma or aspirin sensitivity [86].

## 3. Methods

### 3.1. Study Design and Protocol

In our prospective in vitro study, patients with CRSwNP who underwent surgical polypectomy at our institution Fondazione Policlinico Universitario A. Gemelli IRCCS in Rome, Italy, between January 2018 and June 2022, were included.

All patients gave their written informed consent to participate. The study protocol was approved by the local Ethics Committee of our institution (Number of Protocol: ID 1668).

We screened patients between 18 and 75 years with clinical symptoms: endoscopic evidence of nasal polyposis and sino-nasal occupancy at CT scan of nasal and paranasal cavities, which had been receiving regular treatment with intranasal corticosteroids during the previous 4 weeks, had to be present.

We excluded patients with (1) symptoms of CRS without polyposis, (2) current systemic or upper respiratory tract infections, (3) previous sino-nasal surgery, (4) previous diagnosis of ciliary dyskinesias, cystic fibrosis or nasal malformations, (5) regular treatments of human monoclonal antibodies.

All enrolled subjects underwent otolaryngological and allergological evaluation, including the following procedures.

### 3.2. Skin-Prick Tests (SPTs)

Skin-prick tests were performed, using the main extracts of perennial and seasonal aeroallergens (Lofarma, Milan, Italy), on the volar part of the forearm of each patient at a distance of at least 2–3 cm from the wrist and from the antecubital fold, with a distance between the different extracts of ≥2 cm. For the puncture of the superficial layers of the skin, sterile plastic lancets with a 1 mm unilateral tip were used, which were passed through a drop of the allergenic extract and kept pressed against the skin for at least 1 s with equal pressure applied to each test. Each allergen was punctured with a new lancet to avoid the risk of contamination. A histamine solution (10 mg/mL) and a physiological saline solution (0.9%) were used as positive and negative controls, respectively. Skin prick tests were read 15–20 min after applying the extracts and the reaction was considered positive if it caused the appearance of a wheal ≥ 3 mm in diameter [87].

### 3.3. Endoscopic Evaluation and Tomographic Score

Fiberoptic nasal endoscopy was performed in all subjects under local anesthesia with topical application of 2% xylocaine using 0° and 30°, and a 4 mm diameter rigid nasal endoscope (Karl Storz, Tuttlingen, Germany). Nasal endoscopy was done by using the standard three pass technique as described by Kennedy. Nasal endoscopy findings were recorded using the Lund-Kennedy Endoscopic Scoring system 8, to assess the following parameters: nasal mucosa edema (absent = 0; mild-moderate = 1 or polypoid degeneration = 2), presence of secretion (absent = 0, hyaline = 1 or thick and/or mucopurulent = 2) and presence of polyps (absent = 0, limited to the middle meatus = 1 or extended to the nasal cavity = 2), scarring (absent = 0; mild = 1; severe = 2) and crusting (absent = 0; mild = 1; severe = 2). The assessment was performed bilaterally, with the total points corresponding to the sum of values obtained for both sides. A total score > 4 was considered as indicative of significant modifications of the endoscopic appearance [88].

The Lund-Mackay CT score (range, 0–12 each nostril; higher scores indicate worse status) was used to assess the severity of the disease [89].

### 3.4. Sino-Nasal Outcome Test (SNOT-22)

SNOT-22 is composed of 22 CRS-related items scored from 0 to 5 (total score range 0–110, higher scores represent worse symptoms). SNOT-22 items consist of 2 categories: questions about physical symptoms (items 1–12) and questions about health and QOL (items 13–22) [90,91,92].

### 3.5. Tissue Samples and Cells

Nasal polyp tissue samples were obtained, at the time of polypectomy, from 12 patients, aged 18–70 years. This biological sample was appropriately washed with physiological or PBS, cut into small pieces of 2–3 mm and then plated in 35 mm Petri dishes, containing DMEM medium (Dulbecco’s modified Eagle’s medium, Sigma-Aldrich, St Louis, MO, USA) with addition of 10% fetal bovine serum, 100 g/mL of penicillin/streptamycin, 2 mM L-glutamine, 20 mM of HEPES buffer. The cells were placed in an incubator at 37 °C and 5% CO_2_ [35].

After 7–10 days, the fibroblasts leaked from the portion of nasal polyp and the cells were brought to confluence, trypsinized and expanded into other plates to obtain subcultures, up to the number of cells necessary for the experiment.

Then, fibroblasts were seeded in 96-well plates, into which three drugs were tested, diclofenac, ketoprofen, and lysine acetylsalicylate, at a concentration ranging from 20 to 2000 μg/mL. Furthermore, each condition was realized in quadruplicate. Cell growth was evaluated 2 days after the addition of the drugs and the use of the Cell Counting Kit-8 (CCK-8), (Sigma-Aldrich, St Louis, MO, USA), which gave rise to a colorimetric reaction directly proportional to the number of viable cells present in the well. The absorbance of each well was read by a microplate reader (iMark ™ Microplate Absorbance Reader, Bio-Rad, Hercules, CA, USA) at 450 nm.

### 3.6. Statistical Analysis

Given the purely explorative nature of the study, no a priori hypotheses were formulated; therefore, a formal calculation of the sample size was not performed. A minimum number of 10 patients to be enrolled was defined. The Kolmogorov–Smirnov test was used to evaluate the normality of values. Skewed variables were expressed as median and interquartile range. Comparison between groups was done using the Kruskal–Wallis test. The *p*-value < 0.05 was considered significant. Statistical analysis was performed using Stata version 9 (StataCorp LLC, College Station, TX, USA).

## 4. Results

Thirty patients, suffering from nasal polyposis and candidates for surgery, were screened. During clinical work-up, 18 were excluded as follows: five patients were receiving regular treatment with human monoclonal antibodies. Three patients did not undergo surgery due to hematological comorbidities. Finally, 10 patients did not give written informed consent. The remaining 12 subjects were included in the study.

Their demographic and clinical characteristics are reported in Table 1.

The median values of absorbance fibroblast proliferation cultured in the presence of drugs are presented in Table 2.

The impact of diclofenac, ketoprofen and lysine acetylsalicylate on fibroblast proliferation was similar at concentration of 20 µg/mL (Figure 3A. *p* = 0.8127, Kruskal–Wallis test). The increase of concentration of drugs at 200 µg/mL revealed a trend in favor of diclofenac compared to the other tested drugs, as well as controls (Figure 3B. *p* = 0.1436, Kruskal–Wallis test). Finally, the further increase of concentration of drugs at 2000 µg/mL demonstrated a significant antiproliferative effect of diclofenac compared to other anti-inflammatory drugs (Figure 3C. *p* = 0.0000, Kruskal–Wallis test).

## 5. Discussion

Different approaches to prevent nasal polyp growth are described in the literature: from topical steroids to monoclonal antibody therapy and phytotherapy. The search for alternative strategies was prompted by the frequency of postsurgical recurrences in CRSwNP patients [93]. Various therapeutic strategies are established on the action of pharmacologically active molecules regarding the proliferation of fibroblasts and the control of inflammation of the respiratory mucosa.

INCs are certainly the most-used drugs for their anti-inflammatory and anti-edematous effect. Several in vitro and in vivo studies have shown the key role of INCs in inhibiting fibroblast proliferation by acting as a modulator of inflammation, which reduces basic fibroblast growth factor level in the nasal mucosa [21].

Multiple defects in the innate function of the airway epithelial barrier, including loss of barrier integrity, favor the development of chronic inflammation in CRSwNP [4]. To date, the mechanism that regulates movement of Na^+^ and Cl^−^ across the epithelial cells is considered critical to the pathogenesis of the edema that occurs in nasal polyposis [94,95].

The use of corticosteroids has been shown to modulate epithelial sodium channel and cystic fibrosis transmembrane conductance regulator (CFTR) ion channel expression [96]. Furthermore, INCs seem to act on the aquaporins, reducing edema. In polyp samples from patients treated with fluticasone propionate in the 7 days before sinus surgery, there was a decrease in the expression of aquaporin-2 and aquaporin-5 compared to the control group [97,98].

The use of corticosteroids has been shown to be safe and effective to prevent relapses of nasal polyps prolonging recurrence-free time [27]. The molecules MF, FP, and fluticasone furoate are mainly used, but a comparable efficacy to placebo has been reported in some studies [29,31]. Topical corticosteroids are free from systemic side effects; however, their long-term use may cause nasal irritation and epistaxis.

Because of their anti-inflammatory effects, NSAIDs would be a therapeutic alternative to the use of INCs. Among them, LAS is the most studied and used NSAID and shows good efficacy in several clinical studies regarding patients undergoing surgery [38,40].

An inhibitory effect on the proliferation of fibroblasts has been observed in in vitro studies; however, the mechanisms by which aspirin acts still need to be fully understood [35]. Currently, there are no comparative studies on LAS or other NSAIDs in the literature and, despite the demonstrated clinical efficacy of LAS, the potential use of other NSAIDs is limited.

Currently, no comparative studies on LAS and other NSAIDs are reported in literature and, despite the demonstrated clinical efficacy of LAS, the potential use of other NSAIDs is limited.

For this reason, we aimed to assess the in vitro efficacy of other NSAIDs on nasal tis-sue taken from 12 patients during polypectomy to compare them with LAS. We evaluated the in vitro cell growth of untreated and treated nasal tissue samples with increasing doses of LAS, ketoprofen, or diclofenac. At 20 µg/mL doses, the three NSAIDs used did not demonstrate a statistically significant antiproliferative effect. Furthermore, the three NSAIDs used showed comparable efficacy (*p*-value 0.8127). Increasing the dose to 200 µg/mL, diclofenac was more effective than LAS and ketoprofen (*p*-value 0.1436) in reducing fibroblast proliferation. Additionally, the growth-inhibiting effect of diclofenac at a concentration of 2000 µg/mL was statistically significant compared to other anti-inflammatory drugs (*p*-value 0.0000). LAS and ketoprofen at dosages of 200 and 2000 µg/mL have proven to be effective in reducing fibroblast proliferation with similar results.

These outcomes indicate that diclofenac, even more than ASA, has an important role in inhibiting fibroblast proliferation, and should be considered a valid therapeutic strategy in preventing relapse in patients with CRSwNP.

In the literature, only Ostwald’s group attempted to compare the in vitro efficacy of NSAIDs and INCs. Diclofenac and piroxicam proved to be inferior to mometasone, fluticasone, and beclomethasone but superior to methylprednisolone [41]. However, only concentrations of 30 µg/mL were used. When compared to nimesulide and ibuprofen, which demonstrate no activity, rofecoxib was effective in suppressing fibroblast proliferation in a single study [42].

Regarding pharmacological safety, NSAIDs may provoke side effects in the gastrointestinal system, when administered systemically, and, thus, they may not be suitable for some patients [99]. Topically applied NSAIDs are known to be safer in this regard than oral forms and they might be more effective [100,101].

Some authors searched for alternatives to INCs and NSAIDs based on the known mechanisms of action of other drugs. Furosemide inhibits NKCC2, an isoform of the sodium–potassium–chloride symporter, localized on the basolateral surface of nasal polyp epithelial cells [46]. In one study, it showed similar efficacy to MF in the post-surgical prevention of recurrence [44]. Clarithromycin and mitomycin C could play a role in this process due to their anti-inflammatory effect, by inhibiting the production of pro-inflammatory cytokines, such as IL-5 and GM-CFS [44,49,57]. Antileukotrienes have also demonstrated efficacy comparable to that of INCs in preventing the formation of nasal polyps [102].

Some authors have evaluated the use of capsaicin and medicinal herbs with encouraging results [63,72].

Finally, transcranial electrostimulation and nasal douches with isotonic or hypertonic solutions could also play a role [66,67,68].

However, clinical studies regarding the efficacy of furosemide, antibiotics, antileuko-trienes, and all other alternative approaches are few, and are based on limited patient cohorts. In addition, there is no evidence of the usefulness of antifungals in the post-surgical treatment of nasal polyposis. Finally, biological therapy with monoclonal antibodies showed its effectiveness as an alternative to surgery or in patients with frequent relapses [76,80,86]. However, due to the considerable cost, it is currently impossible to use it to prevent recurrences.

## 6. Conclusions

Stopping recurrences of nasal polyps is a challenge that is far from solved. Several molecules have demonstrated their effectiveness in reducing fibroblast proliferation and nasal polyp formation. Especially, INCs are ideal candidates to intervene in polyp growth through modulation of the inflammation; however, their long-term use can lead to side effects.

Instead, LAS demonstrated good results in vitro and in vivo. Our contribution highlights the role of other NSAIDs, such as diclofenac, in inhibiting fibroblast growth in vitro, which was even more significant than the effect of LAS.

Because only a few studies have reported results of COX-2 inhibitors and other molecules belonging to different groups to reduce fibroblast proliferation, we suggest that further in vitro and in vivo studies are needed.

In conclusion, LAS, ketoprofen, and diclofenac could be valid alternatives to INCs in reducing nasal polyp recurrence.

## Figures and Tables

**Figure 1 jcm-12-02841-f001:**
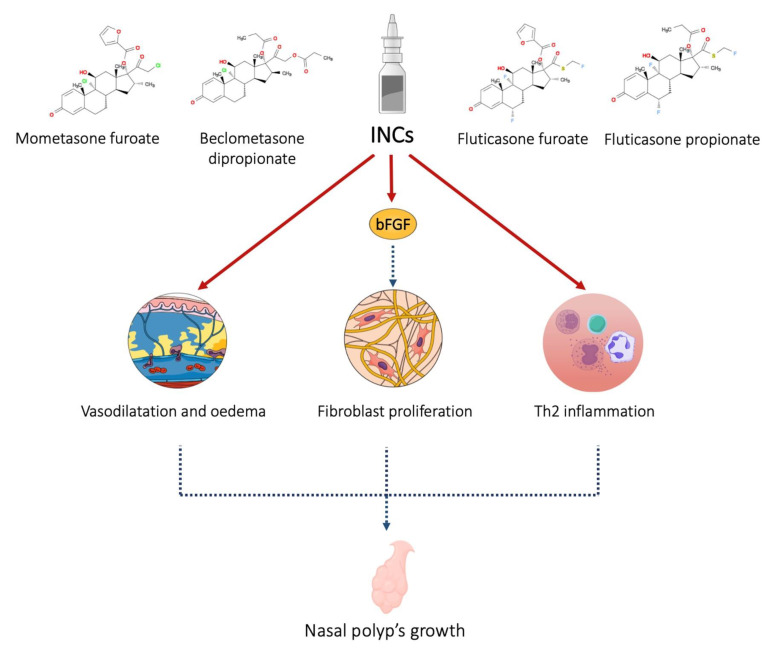
INCs act on the mucosa by reducing vasodilation and edema, inhibiting the production and release of pro-inflammatory cytokines. Furthermore, through the inhibition of basic fibroblast growth factor, they reduce the proliferation of fibroblasts and tissue remodeling.

**Figure 2 jcm-12-02841-f002:**
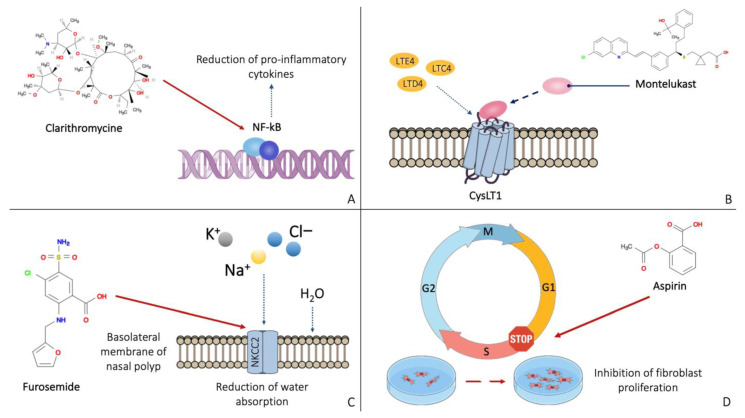
Mechanisms of action of the main molecules active on the nasal polyp: clarithromycin has an anti-inflammatory effect by inhibiting NF-kB (**A**). Montelukast binds the leukotriene receptor CysLT1, preventing its activation (**B**). Furosemide inhibits the NKCC2 receptor on the basolateral membrane of nasal polyp cells (**C**). Aspirin appears to block cells in the G1 phase, preventing the proliferation of fibroblasts (**D**).

**Figure 3 jcm-12-02841-f003:**
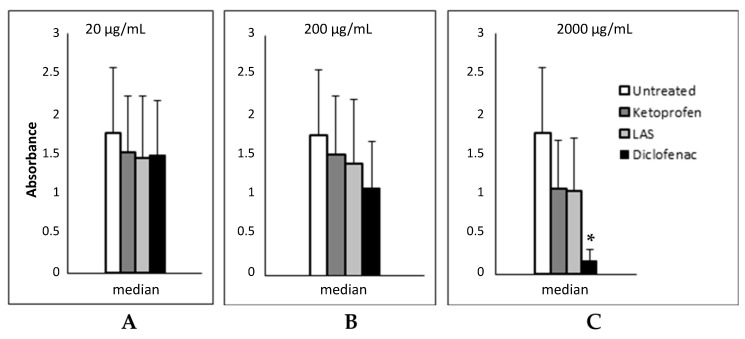
Absorbance values of untreated and exposed cells at concentration of 20 µg/mL (**A**), 200 µg/mL (**B**) and 2000 µg/mL (**C**), respectively. *, *p* = 0.0000, Kruskal–Wallis test.

**Table 1 jcm-12-02841-t001:** Characteristics of enrolled patients.

Patient	Age [Years]	SNOT-22	Nasal Polyp Score (NPS)	Lund-Mackay Score	Perennial Allergen	Seasonal Allergen
#5	60	32	3	10-9	Yes	None
#11	55	33	3	9-8	None	None
#13	53	28	2	7-6	Yes	Yes
#14	69	35	4	12-12	Yes	None
#16	71	36	4	10-11	None	Yes
#17	57	23	2	7-7	None	None
#18	51	26	2	8-7	Yes	Yes
#19	49	25	3	7-6	Yes	None
#20	29	25	3	6-7	Yes	None
#22	55	27	3	8-7	Yes	Yes
#23	53	22	2	7-5	None	None
#25	42	37	4	9-11	None	None

**Table 2 jcm-12-02841-t002:** Absorbance values of untreated and exposed cells.

Patients	#5	#11	#13	#14	#16	#17	#18	#19	#20	#22	#23	#25	Median
**Untreated**	2.3715	1.787	1.202	2.6218	2.3605	2.8755	1.617	1.4245	1.129	0.1373	2.3623	1.2198	1.702
	**20 µg/mL**												
**Diclofenac**	1.845	1.5093	1.1458	2.0838	2.0078	2.7838	1.419	1.267	0.8843	0.167	1.8443	1.0708	1.419
**Ketoprofen**	1.9828	1.539	1.167	2.0848	2.22	2.6737	1.479	1.2525	0.9623	0.1845	2.036	1.0625	1.509
**LAS**	1.9485	0.8665	1.0578	2.195	2.218	2.6933	1.468	1.2598	0.8853	0.0965	2.118	1.0243	1.3639
	**200 µg/mL**												
**Diclofenac**	1.51925	1.1573	0.948	1.5153	1.6033	2.2533	1.0163	0.9938	0.628	0.0583	1.0518	0.82	1.034
**Ketoprofen**	1.9165	1.6093	1.0598	2.0133	2.2838	2.7077	1.4755	1.3095	0.8785	0.1808	1.9455	1.1203	1.5424
**LAS**	1.8888	0.7288	1.143	2.0575	2.217	2.697	1.421	1.172	0.829	0.0933	2.0435	1.058	1.2965
	**2000 µg/mL**												
**Diclofenac**	0.8615	0.5005	0.22	0.1143	0.0858	0.258	0.1633	0.0665	0.1318	0.0125	0.1493	0.1188	0.1405
**Ketoprofen**	1.365	1.2745	0.8313	1.516	1.6878	2.1663	0.8798	0.832	0.5155	0.1133	1.1848	0.7887	1.0323
**LAS**	1.5223	0.5135	1.0483	1.774	1.776	2.0195	0.9043	0.4963	0.7365	0.0403	1.4398	0.7285	0.9763

## Data Availability

The data presented in this study are available on request from the corresponding author.

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
