# Peer review of "Therapeutic Strategies to Prevent the Recurrence of Nasal Polyps after Surgical Treatment: An Update and In Vitro Study on Growth Inhibition of Fibroblasts"

_jcm, 2023, doi:10.3390/jcm12082841_

Round 1
Reviewer 1 Report
This is review paper which describes the postoperative treatment for prevention of nasal polyp. Actually, the exact pathogenesis of the polyp formation has not been completely studied. This paper did good review and their additional in vitro study.
In the discussion section, please add the ion channel mechanism of topical nasal steroid, furosemide and NSAIDs.
Author Response
April 4th, 2023
To Editor and Reviewers
Journal of Clinical Medicine MDPI
We would like to greatly thank the Editor and Reviewers who encouraged a complete revision of the manuscript.
Please find enclosed revision vers. 1 of the Review article entitled “Therapeutic Strategies to Prevent the Recurrence of Nasal Polyps after Surgical Treatment: An Update and In Vitro Study on Growth Inhibition of Fibroblast” by Angela Rizzi, Luca Gammeri, Raffaele Cordiano, Mariagrazia Valentini, Michele Centrone, Sabino Marrone, Riccardo Inchingolo, Franziska Michaela Lohmeyer, Carlo Cavaliere, Francesco Ria, Gabriella Cadoni, Sebastiano Gangemi and Eleonora Nucera.
[Journal of Clinical Medicine] Manuscript ID: jcm-2296873- Major Revisions
Author's Reply to the Review Report (Reviewer 1)
Comments and Suggestions for Authors
This is review paper which describes the postoperative treatment for prevention of nasal polyp. Actually, the exact pathogenesis of the polyp formation has not been completely studied. This paper did good review and their additional in vitro study.
We thank the Reviewer for the comment.
In the discussion section, please add the ion channel mechanism of topical nasal steroid, furosemide and NSAIDs.
We thank the Reviewer for the comment. We modified the discussion section adding evidence of impact of drugs on ion channel.
With the best regards,
Angela Rizzi, Luca Gammeri, Raffaele Cordiano, Mariagrazia Valentini, Michele Centrone, Sabino Marrone, Riccardo Inchingolo, Franziska Michaela Lohmeyer, Carlo Cavaliere, Francesco Ria, Gabriella Cadoni, Sebastiano Gangemi and Eleonora Nucera
Corresponding Author:
Riccardo Inchingolo, MD, PhD
UOC Pneumologia, Fondazione Policlinico Universitario A. Gemelli IRCCS. Largo A. Gemelli, 8 – 00168 – Rome, Italy.
riccardo.inchingolo@policlinicogemelli.it
Corresponding Author will receive all editorial communications.
The authors declare that the manuscript, or specified parts of it, have not been and will not be submitted elsewhere for publication.
Reviewer 2 Report
The authors wrote an article regarding An In Vitro Study on Growth Inhibition of Fibroblast using Nsaids as Therapeutic Strategies to Prevent the Recurrence of Nasal Polyps after Surgical Treatment. They also added a A Literature Review.
The article is very interesting, well written and the topic is hot. I found some problems regarding the second paragraph. The Literature review of molecules used to prevent nasal polyp recurrences is an article in the article. It should be summerized and added in the part of discussion, underlining the new biological therapy and the paragraph of Acetylsalicylic acid and other NSAIDs, eliminating all the rest.
The title should be modified, elminating the abbravation and "literature review".
The manuscript should be considered a brief and preliminary in vitro study and not a review.
In the statistical analysis, please use the kolmorog smirnov test to analyze the normality of values.
Underline the potential topical ans systemic side effects of Nsaids in discussion and coclusion.
Very interesting study, but with the new indications for biological therapy for recurrence, I think that the applicability is very low.
Author Response
April 4th, 2023
To Editor and Reviewers
Journal of Clinical Medicine MDPI
We would like to greatly thank the Editor and Reviewers who encouraged a complete revision of the manuscript.
Please find enclosed revision vers. 1 of the Review article entitled “Therapeutic Strategies to Prevent the Recurrence of Nasal Polyps after Surgical Treatment: An Update and In Vitro Study on Growth Inhibition of Fibroblast” by Angela Rizzi, Luca Gammeri, Raffaele Cordiano, Mariagrazia Valentini, Michele Centrone, Sabino Marrone, Riccardo Inchingolo, Franziska Michaela Lohmeyer, Carlo Cavaliere, Francesco Ria, Gabriella Cadoni, Sebastiano Gangemi and Eleonora Nucera.
[Journal of Clinical Medicine] Manuscript ID: jcm-2296873- Major Revisions
Author's Reply to the Review Report (Reviewer 2)
Comments and Suggestions for Authors
The authors wrote an article regarding An In Vitro Study on Growth Inhibition of Fibroblast using NSAIDs as Therapeutic Strategies to Prevent the Recurrence of Nasal Polyps after Surgical Treatment. They also added a Literature Review.
The article is very interesting, well written and the topic is hot. I found some problems regarding the second paragraph. The Literature review of molecules used to prevent nasal polyp recurrences is an article in the article. It should be summarized and added in the part of discussion, underlining the new biological therapy and the paragraph of Acetylsalicylic acid and other NSAIDs, eliminating all the rest.
We thank the Reviewer for the comment. We first performed a search of published experiences on all drugs with potential effects on fibroblast growth in CRSwNP. The literature of the last 10 years constitutes the rationale for our in vitro study. Therefore, we prefer to maintain the current organization and typology of the manuscript.
The title should be modified, eliminating the abbreviation and "literature review".
We thank the Reviewer for the comment. We removed the abbreviations and replaced “literature review” with “an update”.
The manuscript should be considered a brief and preliminary in vitro study and not a review.
We respect the Reviewer’s comment, but we didn’t agree with him/her. The extensive literature review work is an essential part of the manuscript and is the backbone for subsequent in vitro study. The update according to described criteria covers the last decade of the literature. Therefore, the manuscript maintains the nature of a "review" and includes an original "in vitro study".
In the statistical analysis, please use the kolmorog smirnov test to analyse the normality of values.
We thank the Reviewer for the comment. We added the test used to evaluate the normality of values.
Underline the potential topical and systemic side effects of NSAIDs in discussion and conclusion.
We thank the Reviewer for the comment. We added a paragraph focused on topical and systemic effects of NSAIDs in Discussion section.
Very interesting study, but with the new indications for biological therapy for recurrence, I think that the applicability is very low.
We thank the Reviewer for the comment, but we do not agree on the low applicability of this therapeutic strategy.
Recent literature confirms a global effort by the scientific community towards expanding the therapeutic offer for the disabling reality of chronic rhinosinusitis, especially if complicated/associated with nasal polyposis. Indeed, research of articles published in the last 5 years, using the international database PubMed and keywords “topical NSAIDs” AND “rhinosinusitis” OR “nasal polyposis”, highlights a progressive interest on this topic. Furthermore, one advantage of topical NSAIDs is their ability to strongly inhibit localized inflammation without the potential systemic side effects associated with oral administration.
Future efforts to improve topically applied intranasal NSAIDs’ formulations, such as those demonstrating thixotropic behaviour, allow the drug to maintain a prolonged contact with the nasal lining optimizing local anti-inflammatory effects (doi: 10.26355/eurrev_202212_30486).
Finally, this approach enriches the therapeutic range available for the complex and multidisciplinary management of patients with CRSwNP, in addition to the therapeutic option of new biological drugs.
With the best regards,
Angela Rizzi, Luca Gammeri, Raffaele Cordiano, Mariagrazia Valentini, Michele Centrone, Sabino Marrone, Riccardo Inchingolo, Franziska Michaela Lohmeyer, Carlo Cavaliere, Francesco Ria, Gabriella Cadoni, Sebastiano Gangemi and Eleonora Nucera
Corresponding Author:
Riccardo Inchingolo, MD, PhD
UOC Pneumologia, Fondazione Policlinico Universitario A. Gemelli IRCCS. Largo A. Gemelli, 8 – 00168 – Rome, Italy.
riccardo.inchingolo@policlinicogemelli.it
Corresponding Author will receive all editorial communications.
The authors declare that the manuscript, or specified parts of it, have not been and will not be submitted elsewhere for publication.
Round 2
Reviewer 2 Report
After corrections, the article is ready for publication